# Full-Length Transcriptome Characterization and Functional Analysis of Pathogenesis-Related Proteins in *Lilium* Oriental Hybrid ‘Sorbonne’ Infected with *Botrytis elliptica*

**DOI:** 10.3390/ijms24010425

**Published:** 2022-12-27

**Authors:** Wenting Du, Nan Chai, Zhengqiong Sun, Huiru Wang, Sixian Liu, Shunzhao Sui, Lan Luo, Daofeng Liu

**Affiliations:** 1Chongqing Engineering Research Center for Floriculture, Key Laboratory of Horticulture Science for Southern Mountainous Regions of Ministry of Education, College of Horticulture and Landscape Architecture, Southwest University, Chongqing 400715, China; 2College of Life Sciences, South China Agricultural University, Guangzhou 510642, China; 3Chongqing Institute of Green and Intelligent Technology, Chinese Academy of Sciences, Chongqing 400722, China

**Keywords:** full-length transcriptome, PacBio Sequel II, *Lilium*, *Botrytis elliptica*, PRs

## Abstract

Gray mold (*Botrytis elliptica*) causes a deleterious fungal disease that decreases the ornamental value and yield of lilies. *Lilium* oriental hybrid ‘Sorbonne’ is a variety that is resistant to gray mold. Understanding the mechanism of resistance against *B. elliptica* infection in ‘Sorbonne’ can provide a basis for the genetic improvement in lily plants. In this study, a PacBio Sequel II system was used to sequence the full-length transcriptome of *Lilium* ‘Sorbonne’ after inoculation with *B. elliptica*. A total of 46.64 Gb subreads and 19,102 isoforms with an average length of 1598 bp were obtained. A prediction analysis revealed 263 lncRNAs, and 805 transcription factors, 4478 simple sequence repeats, and 17,752 coding sequences were identified. Pathogenesis-related proteins (PR), which may play important roles in resistance against *B. elliptica* infection, were identified based on the full-length transcriptome data and previously obtained second-generation transcriptome data. Nine non-redundant potential LhSorPR proteins were identified and assigned to two groups that were composed of two LhSorPR4 and seven LhSorPR10 proteins based on their genetic relatedness. The real-time quantitative reverse transcription PCR (qRT-PCR) results showed that the patterns of expression of nine differentially expressed PR genes under *B. elliptica* stress were basically consistent with the results of transcriptome sequencing. The pattern of expression of *LhSorPR4s* and *LhSorPR10s* genes in different tissues was analyzed, and the expression of each gene varied. Furthermore, we verified the function of *LhSorPR4-2* gene in *Lilium*. The expression of *LhSorPR4-2* was induced by phytohormones such as methyl jasmonate, salicylic acid, and ethephon. Moreover, the promoter region of *LhSorPR4-2* was characterized by several functional domains associated with phytohormones and stress response. The overexpression of *LhSorPR4-2* gene in ‘Sorbonne’ increased the resistance of the lily plant to *B. elliptica* and correlated with high chitinase activity. This study provides a full-length transcript database and functionally analyzed the resistance of PR gene to *B. elliptica* in *Lilium*, thereby introducing the candidate gene *LhSorPR4-2* to breed resistance in *Lilium*.

## 1. Introduction

Lily (*Lilium* spp.) is a common ornamental plant in the world which dominates the global commercial floriculture market [1]. Lily can be planted as edible bulbs or for medicinal and ornamental purposes [2,3]. However, the production and economic value of lily are reduced by gray mold, which is a fungal disease caused by *Botrytis* spp. [4]. This disease infects more than 200 types of plants and seriously affects crop yields; prevention and control of its infection are challenging [5]. Gray mold disease is caused by the *Botrytis* species *B. elliptica* and *B. cinerea*. *B. elliptica* is the major pathogen that causes gray mold in *Lilium* [6]. Blight symptoms are primarily observed on the leaves, stems, and flowers during infection, which results in significant losses of yield [7]. The *Lilium* oriental hybrid ‘Sorbonne’ has good resistance to gray mold infection [5,8]. Therefore, ‘Sorbonne’ has high potential as an effective material to analyze the molecular mechanisms that underlie resistance to *B. elliptica* [9].

Plants synthesize diverse proteins, such as pathogenesis-related (PR) proteins in response to infection by various pathogens [10,11]. PR proteins exhibit multiple functions in plants [12]. Although the antibacterial, insecticidal, and antiviral activities of some PR proteins have been reported, most PRs exhibit antifungal activity [13,14,15]. PR proteins are classified into 17 classes based on their differences in structure, physicochemical properties, and biological activities [16]. The PR-4 protein is a chitin-binding protein with high chitinase activity. The PR-4 protein has been previously cloned from various plants such as *Capsicum chinense* [17], rice (*Oryza sativa*) [18], wheat (*Triticum aestivum*) [19] and *Lycoris radiata* [20]. The PR-10 protein family is a multigene family that is present in many higher plants, including monocots [21,22,23] and dicots [24,25,26].

The genome of *Lilium* has not been obtained to date [5,27]. Most previous studies on lily primarily used second-generation sequencing platforms [28,29]. The transcriptome obtained based on the Illumina platform provides rich sequencing information on species whose genomes have not been sequenced, but most of these techniques are not effective at assembling full-length transcripts due to the short sequencing reads obtained [30,31,32]. The PacBio Sequel platform, a third-generation sequencing technology, has ushered in a new era of transcriptome-wide research. This technology is particularly well suited for the direct generation of comprehensive transcriptomes with accurate alternative splicing isoforms and novel genes in non-model organisms that lack genomic sequences. Moreover, advances in single molecule real-time (SMRT) sequencing facilitate the sequencing of high-quality, full-length transcripts, which enables the collection of numerous long-read transcripts with complete coding sequences and the characterization of genes [33,34].

A transcriptome analysis of 18 samples obtained at different stages in the *Lilium* oriental hybrid ‘Sorbonne’ inoculated with *B. elliptica* provides a comprehensive understanding of the defense response of *Lilium*, as well as the potential transcription regulatory network that underlies the defense response against the disease [35]. In this study, the PacBio Sequel II system was used to sequence the full-length transcriptome of ‘Sorbonne’ inoculated with *B. elliptica* using samples collected at different time points after inoculation. An analysis was conducted to explore the *LhSorPR* genes, and their patterns of expression were evaluated. Two highly expressed *LhSorPR4* genes and seven highly expressed *LhSorPR10* genes were identified in this study. Their patterns of expression after *B. elliptica* infection and in different tissues of lily ‘Sorbonne’ were analyzed. Additionally, further functional analysis of the *LhSorPR4-2* gene included the induced pattern of expression after hormone treatments, with the analysis of *cis*-acting elements of this gene promoter, and functional analysis of disease resistance and chitinase activity in the overexpression of the *LhSorPR4-2* gene in transgenic ‘Sorbonne’ plants. These findings provide a useful database for further research in *Lilium* oriental hybrid ‘Sorbonne’ infected with *B. elliptica*, and also provide a comprehensive understanding of its PR genes.

## 2. Results

### 2.1. Sequencing Data Statistics

Full-length transcriptome sequencing was performed on mixed leaf samples harvested from lily ‘Sorbonne’ at different time points after inoculation with *B. elliptica* using the PacBio Sequel sequencing platform (Table 1). The raw sequence data were upload to the NCBI Sequence Read Achieve (SRA) database (https://submit.ncbi.nlm.nih.gov/subs/sra/ (accessed on 14 November 2022)) with accession number: PRJNA914877. A total of 456,489 polymerase reads (51.27 Gb) were obtained with an average length of 112,312 bp and an N50 of 183,827 (Appendix A). The effective insert was produced by a single molecule during sequencing, which represents the sequencing fragment after the removal of the sequencing adapter sequence from the polymerase read and is called a Subread. A total of 44,146,271 Subreads (46.64 Gb) with an average length of 1057 bp and an N50 of 1615 were obtained in this study (Appendix A). Circular consensus sequencing (CCS) provides a sequence with a low error rate that has been sequenced in the same Zero-Mode Waveguide (ZMW) more than twice. The sequencing errors were corrected using multiple sequencing results to ensure high accuracy. A total of 32,349 CCS reads were obtained (Appendix A). The results showed that 253,147 full-length non-chimeric (FLNC) sequences were generated, which accounted for 78.28% of the total number of sequences that had an average length of 1526 bp (Appendix A). FLNC sequences of the same transcript were clustered using the ICE algorithm to obtain the consensus sequence. The resulting consensus sequences were corrected using sequences that did not cover the entire length. A total of 23,801 polished consensus sequences were obtained after clustering. CD-HIT software was further used to cluster the sequences to minimize redundancy, and 19,102 full-length transcript sequences were obtained for subsequent analysis.

### 2.2. Gene Function Annotation

A total of 19,102 isoforms sequences were aligned to the NR, SwissProt, Kyoto Encyclopedia of Genes and Genomes (KEGG) and euKaryotic Orthologous groups (KOG) protein databases using BLASTX with 18,439 sequences (96.53%) aligned to Nr, 16,569 (86.74%) aligned to SwissProt, 14,723 (77.08%) aligned to GO, 9915 (51.91%) aligned to KEGG and 11,700 (61.25%) aligned to the KOG database (Appendix A). The species with the highest number of similar transcripts that were identified using NR annotation included African oil palm (*Elaeis guineensis*), date palm *(Phoenix dactylifera*), and asparagus (*Asparagus officinalis*) with 3845, 1157 and 1015 homologous sequences, respectively (Figure 1A).

In this study, 11,700 transcripts were annotated using the KOG database (Figure 1B). The sequences were grouped into 25 categories according to their functions. “General function prediction only” were the most abundant terms (1950), followed by “Posttranslational modification, protein turnover, chaperones” (1833), whereas the least abundant terms were “Cell motility” (3). The KOG annotation classification indicated that there were 1001 annotations for “Translation, ribosomal structure and biogenesis”, and 978 for “Signal transduction mechanisms”, implying that secondary metabolism and amino acid metabolism were highly active in ‘Sorbonne’.

In addition, 14,723 transcripts were annotated using the GO database (Figure 1C). The results showed that 30,153 genes were involved in biological processes; 40,084 genes were involved in cellular components, and 17,985 genes were involved in molecular functions. An analysis of the cellular fraction terms indicated that there were cells (8708), cell parts (8617), organelle (6173), membrane (4963), membrane parts (6296), whereas the extracellular region parts and nucleoids had 46 and 24 terms, respectively. Biological processes comprised 22 functional groups with a high number of cellular processes (7741), metabolic processes (7212) and single-organism processes (4618), whereas biological adhesion had the lowest number (5).

A total of 9915 transcripts were associated with 127 pathways, as indicated by an analysis of the KEGG database (Figure 1D). The pathways were grouped into five categories, including cellular processing (542), environmental information processing (376), genetic information processing (2288), metabolic processes (6854), and organismal systems (226). The significantly enriched pathways associated with the transcripts included carbohydrate metabolism (1379), energy metabolism (1314), amino acid metabolism (954), and transport and catabolism (542).

### 2.3. Analysis of Gene Structure

The number of transcripts predicted as long non-coding RNAs (lncRNAs) by each software was determined, and Wayne plots were generated. A total of 263 lncRNAs were obtained using a Wayne diagram analysis (Figure 2A).

A total of 17,752 sequences were obtained through CDS prediction analysis (Figure 2B). The sequence length ranged between 276 and 7572 bp with most sequences having a length between 300 and 2000 bp. The length of the 3′ UTR ranged between 0 and 1000 bp, and the length of the 5′ UTR ranged from 0 to 1000 bp (Appendix A). Simple sequence repeats (SSR), or short tandem repeats or microsatellite markers, are repetitive sequences composed of several nucleotides (mononucleotides, dinucleotides, trinucleotides, tetranucleotides, pentanucleotides, and hexanucleotides) as repetitive units.

A total of 4478 SSRs were detected in *Lilium* ‘Sorbonne’, with a high number of mononucleotides (1389, 31.02%), dinucleotides (1269, 28.34%) and trinucleotides (1374, 30.68%). Most of the mononucleotides were composed of 9–12 repeats, while most of the dinucleotides and trinucleotides were 5–8 repeats. Tetranucleotide repeats with 271 sites accounted for 6.05% of the total SSRs. Pentanucleotide repeats had 25 sites that accounted for 0.56% of all the SSRs. Hexanucleotide repeats with 150 sites accounted for 3.35% of the total SSR sites (Figure 2C).

Transcription factors (TFs) play an essential role in transcriptional regulation during plant growth and development, metabolic synthesis, and response to biotic or abiotic stresses. A total of 805 TFs that grouped into 78 families were predicted in this study. The C3H family (53) and C2H2 family (47) had the highest number of TFs, followed by the bHLH family (40), WRKY family (37), and bZIP family (35) (Figure 2D).

### 2.4. Identification of the Lilium Oriental Hybrid ‘Sorbonne’ PR Proteins

A total of 63 differentially expressed genes (DEGs) associated with PR proteins were identified using a previously obtained second-generation transcriptome database [35] and a third-generation transcriptome database (Appendix A). The putative non-redundant PR genes were confirmed by performing BLAST on the NCBI database. The results showed that most of the non-redundant potential candidates were different transcripts of the same genes. Seven distinct *PR10* genes and two different *PR4* genes were identified and denoted as *PR10-1~PR10-7* and *PR4-1~PR4-2* (Appendix A), respectively.

Phylogenetic analyses were conducted using the seven LhSorPR10s and four LhSorPR4s protein sequences. The results showed that seven LhSorPR10s and two LhSorPR4s clustered into two different branches (Figure 3). A phylogenetic analysis showed that LhSorPR10-2, LhSorPR10-3, LhSorPR10-4, LhSorPR10-5, LhSorPR10-1, LhSorPR10-6 and LhSorPR10-7 were grouped in one cluster. LhSorPR4-1 and LhSorPR4-2 formed another cluster. In addition, the length of the open reading frame (ORF) of nine *LhSorPR* genes ranged between 339 (*LhSorPR10-6*) to 474 (*LhSorPR10-2*, *LhSorPR10-3*, and *LhSorPR10-4*) bp, and the lengths of encoded proteins ranged from 112 (LhSorPR10-6) to 181 (LhSorPR10-1) amino acids. The MWs of the proteins ranged from 12301.0 Da to 19790.91 Da, whereas the pI (isoelectric point) values ranged between 4.39 and 10.01.

A heatmap of all the PR genes identified from the DEGs was generated to evaluate the levels of expression of the nine ‘Sorbonne’ genes after inoculation with *B. elliptica* at different times [35]. The levels of expression of genes in the yellow, light red, and dark red modules exhibited fragments per kilobase of transcripts per million mapped reads (FPKM) values at 6, 24, and 48 hpi, respectively, corresponding to the early, middle and late stages of the *B. elliptica* inoculation of ‘Sorbonne’ (Figure 3). Most PR genes showed significant differences in defense responses between the cultivars that were inoculated (AI) and control treatment cultivars (CK). Notably, the *LhSorPR4-2* gene showed a highly significant difference in its level of expression between inoculated and control treatment cultivars at 48 hpi.

### 2.5. Real-Time Quantitative Reverse Transcription PCR (qRT-PCR) Analysis of the LhSorPRs

To further verify the accuracy and reproducibility of the full-length transcriptome sequencing technology, we investigated the profiles of expression of nine *LhSorPR* genes in response to *B. elliptica* infection using qRT-PCR, and the data for gene expression for each *LhSorPR* gene were compared between the after inoculation (AI) and control (CK) tissues at different time points (Figure 4). Within 12 h of *B. elliptica* inoculation, the levels of expression of the other eight genes did not change significantly, except for *LhSorPR10-1*, which was significantly induced after 6 h of pathogen infection in ‘Sorbonne’. Four *LhSorPR* genes were highly expressed until 24 h after inoculation, including *LhSorPR4-1, LhSorPR4-2*, *LhSorPR10-2*, *LhSorPR10-7*, that were 17.3-, 22.7-, 8.6- and 3.1-fold significantly higher than that in the control, respectively, and continued to increase at 36 h. After 36 h of *B. elliptica* inoculation, *LhSorPR10-1* was 3.5-fold significantly higher than that in the control. In addition, at 48 h, *LhSorPR4-1*, *LhSorPR4-2*, *LhSorPR10-1, LhSorPR10-2, LhSorPR10-5* and *LhSorPR10-7* were 60.8-, 61.6-, 5.9-, 12.0-, 1.6-, and 3.7-fold significantly higher than those of the control, respectively. These results were partially consistent with the transcriptome sequencing analysis. Two *PR4* genes and the *PR10* genes except for *LhSorPR10-3*, *LhSorPR10-4* and *LhSorPR10-6* were significantly induced at different times after inoculation with *B. elliptica* in *Lilium*.

Moreover, the profiles of expression of the *LhSorPR4* and *LhSorPR10* genes in different tissues of the *Lilium* oriental hybrid ‘Sorbonne’, including the roots, stems, scales, leaves, and flowers, were evaluated by qRT-PCR (Figure 5). *LhSorPR4-1* was highly expressed in the flowers, while *LhSorPR4-2* showed its highest levels of expression in the stems and secondly in the leaves and flowers. *LhSorPR10-1* was expressed the most highly in roots and secondly in the scales. There was almost no expression in the flowers, stems, and leaves. *LhSorPR10-2* was highly expressed in the leaves, and there was almost no expression in the other tissues. *LhSorPR10-3* was expressed in all the tissues and highly expressed in the flowers and stems. *LhSorPR10-4* was highly expressed in the roots and leaves. *LhSorPR10-5*, *LhSorPR10-6* and *LhSorPR10-7* were all expressed the most highly in the scales compared with the other tissues, and secondly expressed more highly in the roots. These results indicated that the pattern of expression of the *LhSorPR4s* and *LhSorPR10s* are varied and tissue-specific.

### 2.6. Hormone-Induced Pattern of Expression of the LhSorPR4-2 Gene and Analysis of Its Promoter

The results indicated that *LhSorPR4-2* was significantly upregulated from 24 h to 48 h after inoculation with *B. elliptica*. We chose *LhSorPR4-2* for further functional analysis. Salicylic acid (SA), methyl jasmonate (MeJA), and ethephon (ETH) are essential signaling molecules in plant biotic stress, and the signaling pathways that they are involved in have been extensively investigated [24,36]. Therefore, we analyzed the hormone-induced pattern of expression of *LhSorPR4-2* under these three pathogen-associated phytohormone treatments (Figure 6). At 1 h, treatment with SA dramatically increased the expression of the *LhSorPR4-2* gene with a 7.69-fold increase in expression compared with the control. *LhSorPR4-2* was the most highly expressed at 48 h and 6 h after treatment with MeJA and ETH, which exhibited 5.6- and 2.5-fold levels of expression compared with that of the control, respectively. In addition, the *LhSorPR4-2* promoter was amplified from the ‘Sorbonne’ genomic DNA by PCR walking to a length of 1575 bp. An analysis of the *cis*-regulatory elements of the *LhSorPR4-2* promoter sequence revealed that the region comprised 23 elements. The promoter sequence was characterized by the two transcriptional essential elements TATA-Box and CAAT-Box, five light-responsive elements, including the GT1-motif, Box 4, LAMP-element, chs-CMA1a and MRE, and the *cis*-regulatory elements that are essential for anaerobic induction (ARE). In addition, it had three hormone-related elements, including the gibberellin response element P-box, the *cis*-acting TCA-element that is involved in the response to salicylic acid, and the MYB element that responds to dehydration and abscisic acid (ABA) signals. Moreover, the analysis showed the presence of multiple *cis*-acting elements associated with stress, such as the MYBHv1 binding site, W box element, MYC-specific recognition site, and MYB recognition site (Table 2).

### 2.7. Overexpression of LhSorPR4-2 in Lily Enhances Resistance to B. elliptica

To evaluate the role of *LhSorPR4-2* resistance to *B. elliptica*, we constructed a plant vector in which the ORF was modulated by the ZmUbi2 35S promoter and then introduced into the embryogenic callus of *Lilium* oriental hybrid ‘Sorbonne’.

The genetic transformation process of embryogenic callus of overexpressing plants is composed of preincubation, coculture, culture screening, survival seedlings, and rooted seedlings (Figure 7A). The transgenic plants were confirmed by PCR (Appendix A). Overexpressed (OE) transgenic lines (OE1–4) with higher levels of expression were selected for subsequent analyses (Appendix A).

Agar plugs that contained *B. elliptica* mycelia were inoculated onto the leaves of transgenic and wild type (WT) lines to evaluate the resistance of plants that overexpressed the *LhSorPR4-2* gene to *B. elliptica* (Figure 7B). The results showed that the OE plants took a long time to develop disease, and the lesion area was significantly smaller (*p* < 0.01) compared with the WT lines (Figure 7C). In addition, the chitinase activity in the WT and *LhSorPR4-2* OE plants inoculated with *B. elliptica* and water for 72 h was evaluated. The chitinase activity in the plants that overexpressed *LhSorPR4-2* inoculated with water was 0.629 U · g^−1^, which was significantly higher (13% higher) compared with that in the WT before inoculation. After inoculation with *B. elliptica*, the chitinase activity in the *LhSorPR4-2* lily plants was 0.839 U · g^−1^, which was significantly higher (32.09%) compared with that in the WT. These results indicated that the overexpression of *LhSorPR4-2* gene increased the chitinase activity in *Lilium* plants (Figure 7D). In summary, the results indicate that the overexpression of *LhSorPR4-2* gene enhances the resistance of ‘Sorbonne’ to *B. elliptica*.

## 3. Discussion

Gray mold is a deleterious fungal disease that affects the leaves, flowers, and stems of different flowering plants, including the lily plant [6]. *Lilium* oriental hybrid ‘Sorbonne’ is highly resistant to gray mold [5]. Elucidation of the defense mechanism of ‘Sorbonne’ against *B. elliptica* infection will provide information to breed lily plants with resistance traits to gray mold. The third-generation sequencing technique PacBio Iso-Seq is used to study species whose genome has not been obtained, such as *Lilium* species [37]. Full-length transcriptome sequencing, also known as third-generation sequencing, is widely used in current research and has several methodological advantages [38,39]. This technique does not require library assembly and is used to directly obtain high-quality transcriptome information from the 5′ UTR to the 3′ UTR. In addition, this method allows accurate determination of information, such as alternative splicing, variable polyadenylation, fusion genes, gene families, and the non-coding RNAs of genes [34].

In this study, the FLNC sequences accounted for 78.28% of the full-length transcriptome of ‘Sorbonne’. This indicates a high quality of the SMRT data; thus, the data were used to clone the full-length genes, characterize gene function analysis, and perform further in-depth analyses [40]. A total of 19,102 isoforms were obtained in this study. In addition, 96.62% of the isoforms were successfully annotated, which provides an important reference to predict the full-length transcripts of *Lilium* after infection with gray mold [35]. Functional annotations and classifications of the transcripts provided information on changes in the transcriptome of *Lilium* after infection with *B. elliptica*. LncRNAs are non-coding RNAs with no protein-coding function, but are involved in various phases of plant growth and development [41]. In this study, 263 lncRNAs were obtained that can be used to study the mechanism of activities of gray mold infection in *Lilium* in more detail. SSR analysis was conducted for the full-length transcriptome of ‘Sorbonne’ and 4478 SSR loci were identified, which can be used to analyze the genetic diversity of *Lilium* and identify germplasm resources. Moreover, a total of 805 possible TFs were identified, which help to investigate the regulation of gene function, synthesis of active compounds, and metabolic regulation of *Lilium* against *B. elliptica* through prediction analysis and classification of the TFs in full-length transcriptome.

PR proteins are core components of plant defense systems and have been extensively studied in previous research. PR proteins have multiple functions in plants [11]. PR genes have been identified in several plants, such as soybean (*Glycine max*) [42], tea (*Camellia sinensis*) [43], Chinese mustard (*Brassica juncea*) [44], grape (*Vitis vinifera*) [26], European plum (*Prunus domestica*) [24], and ginseng (*Panax ginseng*) [25]. In this study, nine PR genes with complete PR domains were identified and assigned to the *LhSorPR4* and *LhSorPR10* families through transcriptome analysis and sequence alignment. The levels of expression of most of the genes were significantly different between the inoculated and control treatments. This finding indicates that these genes were implicated in the defense response to gray mold infection and should be studied in more detail.

To further verify the accuracy and reproducibility of full-length transcriptome sequencing technology, nine PR genes that may be involved in the disease resistance response of ‘Sorbonne’ were selected from the differentially co-expressed genes identified by transcriptome sequencing for qRT-PCR validation. PR proteins play essential roles in response to pathogen infection in various plants [45]. For example, the expression of PR4 proteins in rice is induced by pathogen attack, as well as by fungal elicitors and chemicals [46]. In this study, the level of expression of most of the *LhSorPR* genes was significantly induced after *B. elliptica* infection compared with the control plants, which was partially consistent with the results of transcriptome sequencing. Tissue-specific patterns of expression of the *LhSorPRs* provide information on the physiological function of the genes. A qRT-PCR revealed that *LhSorPRs* had variable levels of mRNA in all the tissues of *Lilium* oriental hybrid ‘Sorbonne’ examined, which could be related to their function of resistance to disease in stems, leaves, or scales [12].

Phytohormones play crucial roles in plant growth and development, as well as stress tolerance [11]. In this study, the expression of *LhSorPR4-2* was induced by the administration of SA, MeJA, and ETH. SA significantly upregulated the expression of *LhSorPR4-2* at 1 h after treatment, whereas the induction effects of MeJA and ETH were observed at later time points, which indicate that *LhSorPR4-2* plays a positive role in the stress response against *B. elliptica* and the possible defense activities may be primarily modulated by the hormone signaling pathways [47]. In addition, the level of gene expression is controlled by the promoter sequences and corresponding transcription factors during the response to biotic and abiotic stresses [48]. The *LhSorPR4-2* promoter comprises hormone-responsive elements (P-box, TCA-element and MYB), as well as the stress response elements STRE, W-box, MYB-binding site, MYC and WRE3. The presence of these *cis*-acting elements indicated that the expression of *LhSorPR4-2* could be modulated by light signals, phytohormones, and stress. This regulatory profile is analogous to the profile of PR genes in garlic (*Allium sativum*) (*AsPRs*) [49]. Moreover, the presence of stress response elements and hormone-responsive elements explains the induced expression of *LhSorPR4-2* gene after *B. elliptica* infection and hormone treatments, such as SA, MeJA and ETH.

Several studies have explored the biological role of PR-4 in plant defense systems [50,51,52]. A PR-4 protein (VpPR4-1) from a wild Chinese grape (*Vitis pseudoreticulata*) was overexpressed in cultivated grape (*V. vinifera*), which enhanced its resistance to powdery mildew compared with the WT vines [53]. A cDNA fragment (*FaPR4*) was obtained from *Ficus awkeotsang* and overexpressed in the yeast strain *Pichia pastoris*, which enhance the RNase and antifungal activity [54]. *CaHaPR-4* from chickpea (*Cicer arietinum*) exhibits antifungal activity against the growing hyphae of *Fusarium oxysporum* [55]. The MdPR-4 protein, which was isolated from apple (*Malus domestica*), is involved in the defense responses of apple to pathogenic attack by directly inhibiting hyphal growth [56]. In this study, the overexpression of *LhSorPR4-2* gene in *Lilium* oriental hybrid ‘Sorbonne’ delayed the time of onset of infection compared with that of WT plants, and it significantly inhibited *B. elliptica* infection. An increase in the activity of chitinase indicated that *LhSorPR4-2* is an important gene in the defense system of lily plants.

## 4. Materials and Methods

### 4.1. Plant Materials and Pathogen Inoculation

The *Lilium* oriental hybrid cultivar ‘Sorbonne’ used in this study was purchased from Van den Bos Flowerbulbs (Netherlands). The plants were grown in a chamber at a temperature of 25 °C for 16 h during the day and 22 °C for 8 h at night and 50–70% relative humidity. *Botrytis elliptica* strain 36,423 was obtained from the China Agricultural Strain Collection (http://www.accc.org.cn/ (accessed on 8 December 2020)) for inoculation into ‘Sorbonne’ cultivars. The *B. elliptica* strain was isolated from symptomatic lily plants. Fungal mycelia were grown at 25 °C under dark conditions for a week. Potato dextrose agar (PDA; pH 5.8; Coolaber, China) medium in Petri dishes with a 9 cm diameter was used to grow the mycelia. A sterilized puncher was used to take *B. elliptica* mycelial discs (5 mm in diameter) from the PDA plates and inoculate in vitro lily leaves with them. The leaves were obtained at 0, 6, 12, 24, 36, and 48 h post-inoculation (hpi) and weighed. The weights of leaves collected at different time points were approximately equal. Leaf samples collected at different time points after inoculation and pooled together were used in subsequent PacBio Iso-seq sequencing.

### 4.2. RNA Isolation, Library Preparation and Sequencing

Total RNA was extracted from the leaf samples. A Nanodrop 2000 (Thermo Fisher Scientific, Austin, TX, USA) was used to determine the concentration and purity of the extracted RNA. Subsequently, 1% agarose gel electrophoresis was conducted to evaluate the genomic contamination, purity, and RNA integrity of the samples. An Agilent 2100 (Agilent Technologies, Palo Alto, CA, USA) was used to determine the RNA integrity number (RIN) value. The full-length cDNA of the extracted mRNA was synthesized using a Clontech SMARTer PCR cDNA Synthesis Kit (Clontech Laboratories, Frederick, MD, USA). Primers with Oligo dT were used to pair with the A-T bases at the polyA region for the reverse transcription of the mRNA to obtain cDNA. The full-length cDNA was amplified by PCR, and the product was purified using PB magnetic beads to remove fragments of cDNA that were less than 1 kb. The end of cDNA was repaired, and the fragments were connected with the SMRT dumbbell connector. The unconnected fragments were digested using exonuclease and purified using PB magnetic beads to obtain the sequencing library. A Qubit 3.0 fluorometer (Invitrogen, Carlsbad, CA, USA) was used to accurately quantify the library, and an Agilent 2100 was used to determine the library size. The DNA was sequenced after the library size met the criteria.

### 4.3. De Novo Assembly and Gene Function Annotation

Raw sequencing data were preprocessed using SMRTlink (V8.0) software. Complete transcript sequences were obtained through the Iso-Seq analysis process. LoRDEC (V0.9) error correction software was used to correct FLNC sequence data using previously obtained second generation transcriptome data [35], and the correction effect was evaluated. Full-length transcript sequences were obtained by clustering to eliminate redundancy using CD-HIT software [57]. Complete transcript sequences were used in the subsequent analysis as reference transcript sequences for second generation data comparisons. Diamond software was used for molecular visualization [58]. BLASTX was used to retrieve the isoform sequences using the non-redundant (NR) database. GO [59] and KEGG [60] were used for functional analyses. KOG was used to identify orthologs and paralogs of the sequences [61]. Swiss-Prot [62] databases were used to identify proteins with high sequence similarity to obtain the functional annotation information of the sequenced proteins.

### 4.4. Gene Structure Analysis

Most of the full-length sequences in the transcriptome data were well-annotated through the alignment of sequences using five databases. Subsequently, lncRNA analysis was performed for full-length sequences that were not annotated. The coding-non-coding index (CNCI) [63], coding potential calculator (CPC2) [64], PLEK [65] and coding-potential assessment tool (CPAT) [66] tools were used to predict the coding potential of unannotated full-length transcripts. The common regions were selected for prediction of the lncRNAs. Transdecoder (V5.5.0) software was used to predict the CDS regions. MISA (V1.0) was used to identify microsatellites in the full-length transcripts. Diamond (V0.9.24)/iTAK (V1.7) software was used to predict the TFs by comparing the sequences to a TF in the database.

### 4.5. Identification of the PR Genes of Lilium Oriental Hybrid ‘Sorbonne’ after Infection with B. elliptica

The sequences of all the putative PR genes were obtained from the previously obtained second-generation transcriptome data of the oriental lily hybrid ‘Sorbonne’ inoculated with *B. elliptica.* The original reads of ‘Sorbonne’ were uploaded in the Sequence Read Archive (SRA) of the NCBI database (accession number: PRJNA742853) [35]. Highly expressed PR genes were obtained from the second-generation transcriptome and combined with third-generation transcriptome data to obtain full-length sequences. Short *LhSorPRs* sequences with incomplete domains were deleted from the transcriptome data. Predicted *LhSorPRs* with intact domain structures were subjected to BLAST multiple alignment analysis to eliminate duplications. The *LhSorPR* domains were aligned using the Clustal W tool with default settings. MEGA 7.0 was used to infer the phylogenetic relationships of the homologous sequences using the neighbor-joining algorithm. An analysis of the phylogenetic tree was conducted using 1000 iterations to calculate the bootstrap values. The number of amino acids, molecular weight (MW), and theoretical pI(isoelectric point) were evaluated using the ProtParam (https://web.expasy.org/protparam/ (accessed on 26 April 2020)) tool using the nine full-length predicted *LhSorPR* sequences to characterize the *B. elliptica*-responsive LhSorPRs proteins. Subcellular localizations of the proteins were predicted using the ProtComp 9.0 tool. An expression heatmap was generated using the OmicStudio webserver (https://www.omicstudio.cn/tool/4 (accessed on 26 April 2020)).

### 4.6. qRT-PCR

To verify the accuracy of transcriptome sequencing results, nine selected differentially co-expressed *LhSorPR* genes were validated using fluorescence quantitative PCR technology. *B. elliptica* was inoculated into detached leaves from *Lilium* plants at the flower bud stage, and pure water was used as a control. Samples of the infected parts of the leaves were collected at 6 h, 12 h, 24 h, 36 h, and 48 h after inoculation. Three biological replicates were obtained from each sample to analyze the expression of genes after pathogen infection.

In addition, we analyzed the expression of the *LhSorPR4-2* gene following treatment with MeJA, SA, and ETH. ‘Sorbonne’ tissue culture seedlings with a seedling height of 5 cm and a bulb height of 0.5 cm were selected and incubated in sterile water for 24 h for subsequent phytohormone treatments. The seedlings were then transferred to 100 mmol· L^−1^ MeJA, 200 mmol· L^−1^ SA, or 1 mmol· L^−1^ ETH and incubated for 30 min. Samples of the root tissues of cultured seedlings were obtained at 0 h, 1 h, 3 h, 6 h, 12 h, 24 h, and 48 h after hormone treatment.

The levels of expression of actin (ACT) and elongation factor 1 (EF1) were used as internal controls to normalize the levels of expression of PR genes in the RNA samples [27]. SsoFast EvaGreen Supermix (Bio-Rad, USA) was used for the qRT-PCR analysis. The qRT-PCR reaction was performed on a CFX96 Real-Time PCR Detection System (Bio-Rad). The program of the PCR reaction was as follows: 95 °C for 30 s, followed by 40 cycles at 95 °C for 5 s and 58 °C for 5 s. The qRT-PCR primers used in this study are presented in Appendix A.

### 4.7. Cloning and Analysis of the LhSorPR4-2 Promoter

The upstream region of the *LhSorPR4-2* gene was isolated from the *Lilium* oriental hybrid ‘Sorbonne’ genome using a KX Genome Walking Kit (ZOMANBIO, Tianjin, China) [67]. *LhSorPR4-2* gene-specific primers SP1, SP2, SP3 (Appendix A), and the adaptor primers ZFP8, ZSP1, and ZSP2 were used to amplify the *LhSorPR4-2* promoter region [68]. Upstream *cis* elements in the *LhSorPR4-2* promoter were analyzed using the PlantCARE database [69]. The promoter elements were visualized using TBtools v1.09876 tool [70].

### 4.8. Lilium Oriental Hybrid ‘Sorbonne’ Transformation and Analysis of the Disease Resistance of Transgenic Plants

The ORF of *LhSorPR4-2* was amplified using *LhSorPR4-2* F/R primers (Appendix A). The *LhSorPR4-2* ORF was inserted into PVM01-GFP to obtain *LhSorPR4-2* OE vectors using PVM01-*LhSorPR4-2* F/R primers (Appendix A). The vectors were transferred to *Agrobacterium tumefaciens* strain EHA105 to transform the embryogenic callus of ‘Sorbonne’ by electroporation. Transformants were selected using 2.5 mg·L^−1^ phosphinothricin (PPT). The genetic transformation process of embryogenic callus of overexpressing plants comprised preincubation, coculture, culture screening, survival seedlings, and rooted seedlings. The medium used to pre-culture for 10 days was MS + PIC (picloram) 1.0 mg·L^−1^ + sucrose 60 g·L^−1^ + agar 8 g·L^−1^. The culture medium was MS + PIC 1.0 mg·L^−1^ + sucrose 30 g·L^−1^ + MES (2-morpholinoethanesulfonic acid) 10 mM + AS (acetosyringone) 100 μM. A volume of 5 mL was pipetted onto three layers of filter paper and co-cultured in the dark at 25 °C for 3 days. The selection medium was MS + PIC 1.0 mg·L^−1^ + Cb (carbenicillin) 500 mg·L^−1^ + PPT 2.5 mg·L^−1^ + sucrose 30 g·L^−1^ + agar 8 g·L^−1^, and the selection medium was replaced every 15 days. The plant material was grown in the greenhouse under the following conditions: light intensity of 20,000 lx, photoperiod of 16 h light and 8 h dark, 25 °C day/20 °C night, and 85% humidity.

The DNA extracted from the transgenic seedlings obtained was subjected to PCR to confirm its identity.Upstream primers were designed on the gene sequence and downstream on the vector sequence for PCR assays (Appendix A). The plasmid that contained the *LhSorPR4-2* gene was used as a positive control. The untransformed plants served as a negative control. The results of gel electrophoresis of the PCR products are shown in (Appendix A). A qRT-PCR was conducted to evaluate the level of expression of *LhSorPR4-2* in overexpressing plants. A uniform agar plug with hyphae was applied to the surface of each in vitro leaf of the transgenic and wild-type plants to test the resistance of the cultivars to *B. elliptica*. The size of the lesions was determined, and the lesions were photographed 72 h after application of the agar plug. The chitinase activity of transgenic lilies and WT was evaluated after inoculation. A chitinase assay kit (Solarbio Science & Technology Co., Ltd., Beijing, China) was used to detect chitinase activity [71]. Chitinase hydrolyzes chitin to produce N-acetylglucosamine, which further produces a reddish-brown compound with 3,5-dinitrosalicylic acid. The product of this reaction has a characteristic absorption peak at 540 nm, and an increase in the rate of absorption of this peak reflects the activity of chitinase.

## 5. Conclusions

The full-length transcriptome sequencing of the ‘Sorbonne’ inoculation with *B. elliptica* was conducted using the PacBio Sequel II system. In the study, 46.64 Gb subreads and 19,102 isoforms were obtained. These findings lay a foundation to clone the full-length genes, analyze their function, and elucidate the mechanism of synthesis of the active components against *B. elliptica* in ‘Sorbonne’. All the PR genes that responded to the *B. elliptica* infection were identified based on the full-length transcriptome data obtained in this study and the previously obtained second-generation transcriptome data from our previous study. Their patterns of expression were evaluated. We found that the patterns of expression of nine differentially expressed PR genes under *B. elliptica* stress were basically consistent with the transcriptome sequencing results, and the patterns of expression of each gene varied in the different tissues. In addition, a functional analysis of the *LhSorPR4-2* gene was performed. The *LhSorPR4-2* gene showed that high levels of expression were induced by phytohormones, such as MeJA, SA and ETH. The results showed that the promoter region of *LhSorPR4-2* comprised several functional domains implicated in light signaling, phytohormones, and stress responses. Moreover, overexpression of the *LhSorPR4-2* gene in lily resulted in increased resistance against *B. elliptica* and was associated with high chitinase activity. In summary, these findings provide full-length transcriptional data to understand the resistance to *B. elliptica* in *Lilium*, as well as details on PR genes involved in *B. elliptica* infection in the oriental lily hybrid ‘Sorbonne.’ In addition, potential candidate genes to breed varieties of lily that are resistant to *B. elliptica* were identified in this study.

## Figures and Tables

**Figure 1 ijms-24-00425-f001:**
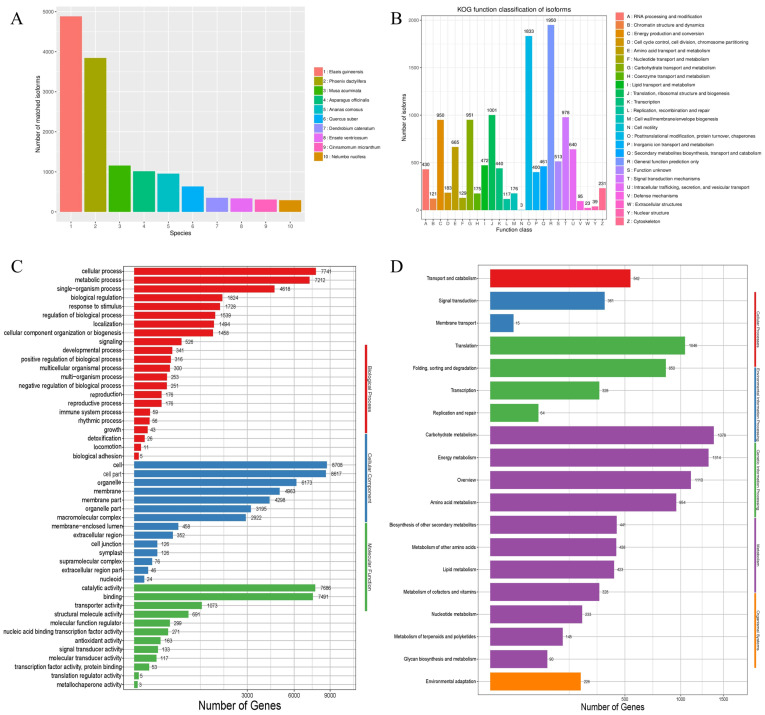
Species distribution annotated from the NCBI NR database (**A**), Functional annotation using the KOG (**B**), functional annotation using the GO database (**C**), and KEGG databases (**D**). GO, Gene Ontology; KEGG, Kyoto Encyclopedia of Genes and Genomes; KOG, euKaryotic Orthologous Groups.

**Figure 2 ijms-24-00425-f002:**
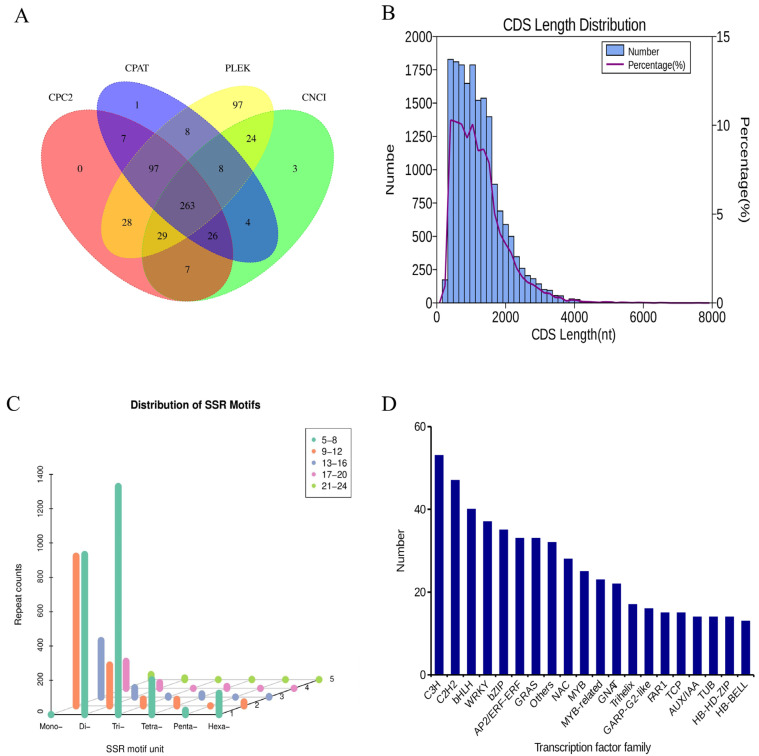
Long non-coding RNA (lncRNA) prediction (**A**), Length distribution of protein-coding sequences (CDSs) (**B**), Single sequence repeat (SSR) identification (**C**), transcription factor (TF) family statistics (**D**). LncRNAs were predicted using CNCI, Pfam, PLEK, and CPC2 computational approaches.

**Figure 3 ijms-24-00425-f003:**
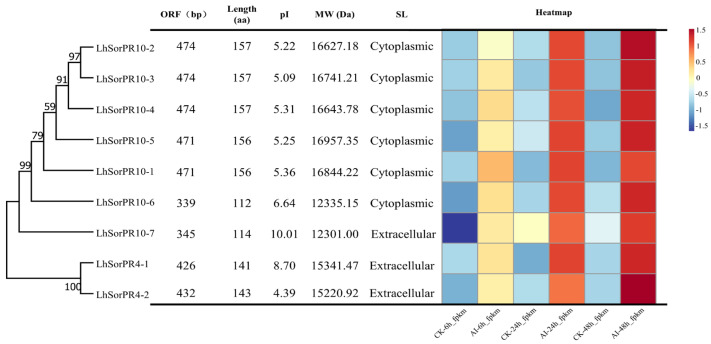
Analyses of the structure and expression of *LhSorPR* genes. From left to right, they are the phylogenetic relationships, protein sequence analysis and hierarchical clustering of gene-expression data of nine *LhSorPR* genes. The phylogenetic tree was constructed by MEGA 7.0. The number of amino acids, molecular weight (MW), and theoretical pI (isoelectric point) were evaluated using the ProtParam tool. A heat map between the AI and CK treatments at three stages of infection. The horizontal direction indicates expression at 6 h, 24 h and 48 h for the treatment and control. The longitudinal direction indicates nine *LhSorPR* genes. The data were FPKM values extracted from previously obtained second-generation transcriptome data (log2 transformed) and the color scale as shown on the right. AI, after inoculation; CK, control; FPKM, fragments per kilobase of transcripts per million mapped reads.

**Figure 4 ijms-24-00425-f004:**
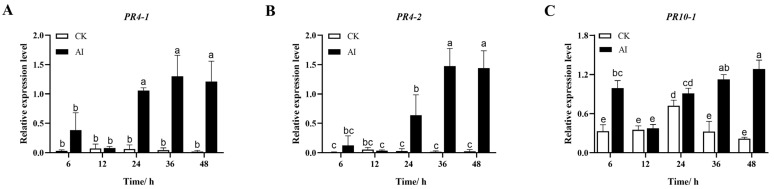
(**A–I**) are a consecutive series of pictures showing time course of the level of the *LhSorPR* genes in ‘Sorbonne’ after infection with *Botrytis elliptica.* Data represent the mean of three biological repeats ± SD. Error bars indicate the standard deviation, and the different lowercase letters on the bars indicate the significant differences (*p* < 0.05).

**Figure 5 ijms-24-00425-f005:**
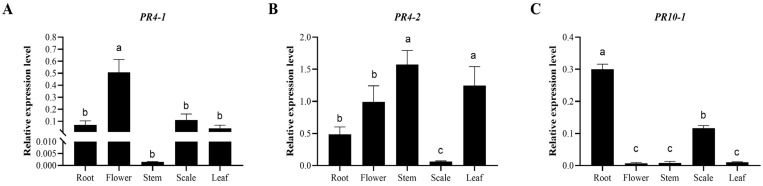
(**A–I**) are a consecutive series of pictures showing levels of expression of *LhSorPRs* in various lily tissues. Data represent the mean of three biological repeats ± SD. Error bars indicate the standard deviation, and the different lowercase letters (a, b, c) on the bars indicate significant differences (*p* < 0.05).

**Figure 6 ijms-24-00425-f006:**
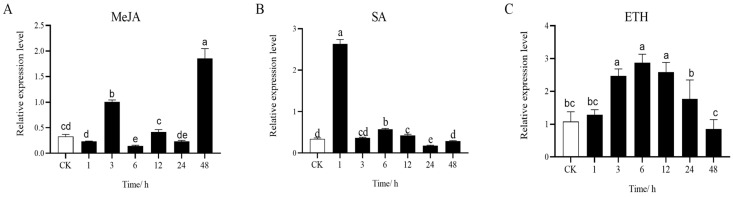
Levels of expression of *LhSorPR4-2* after treatment with MeJA (**A**), SA (**B**) and ETH (**C**). Data represent the mean of three biological repeats ± SD. Error bars indicate the standard deviation, and the different lowercase letters (a, b, c, d, e) on the bars indicate the significant differences (*p* < 0.05). ETH, Ethephon; MeJA, methyl jasmonate; SA, salicylic acid; SD, standard deviation.

**Figure 7 ijms-24-00425-f007:**
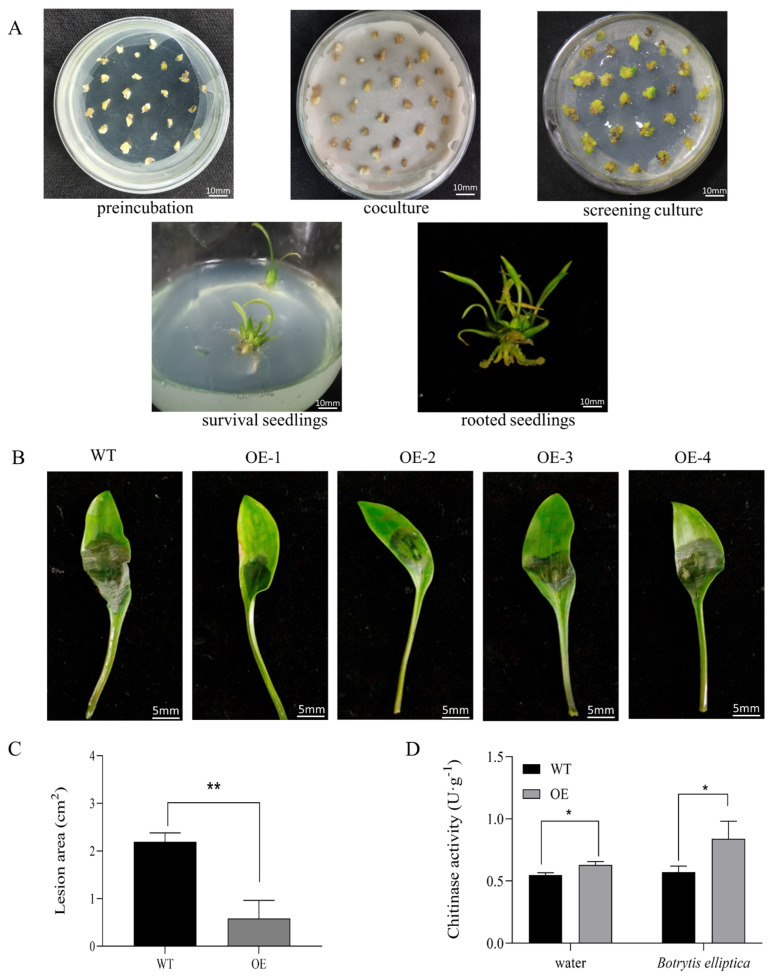
Effect of the overexpression of *LhSorbPR4-2* on the resistance of lily leaves to *Botrytis elliptica* infection. (**A**) The genetic transformation process of lily ‘Sorbonne’ embryogenic callus. (**B**) Images of the leaves from transgenic and wild type (WT) plants infected with *B. elliptica*. OE1, OE2, OE3, and OE4 represent four *LhSorPR4-2* transgenic lines. The leaves were photographed at 72 h after inoculation. (**C**) The average diameters of lesions of infected leaves at 72 h. (**D**) Chitinase activities in lily plants inoculated with *B. elliptica*. Error bars represent the standard errors of three biological replicates. Asterisks indicate statistically significant differences between the transgenic lines and the WT plants (* 0.01 < *p*< 0.05, ** *p* <0.01, Student’s *t*-test).

**Table 1 ijms-24-00425-t001:** Statistics of the SMRT sequencing data.

Category	Total_Bases (Gbp)	Amount	Amount Min Length/bp	Mean Length/bp	Max Length/bp	N50 Length/bp
Polymerase reads	51.27	456,489	51	112,312	329,895	183,827
Subread	46.64	44,146,271	51	1057	264,517	1615
CCS	550,727,214	323,349	53	1704	14,722	2000
FLNC	386,168,108	23,801	134	1608	8231	1853
Polished consensus	38,263,150	23,801	134	1608	8231	1853
Corrected consensus	30,515,381	19,102	134	1598	8231	1859

Annotation: CCS, circular consensus sequencing; FLNC, full-length non-chimeric; SMRT, single molecule real-time.

**Table 2 ijms-24-00425-t002:** *Cis*-acting regulatory element analysis of the *LhSorPR4-2* promoter sequence.

No.	Site Name	Amount	Sequence (5′-3′)	Function of Site
1	CCAAT-box	1	CAACGG	MYBHv1 binding site
2	MYB	2	CAACCA/CAACAG	responds to dehydration and ABA signals
3	TCA	1	TCATCTTCAT	Unknown function
4	P-box	1	CCTTTTG	gibberellin-responsive element
5	ARE	3	AAACCA	*cis*-acting regulatory element essential for the anaerobic induction
6	TCA-element	2	CCATCTTTTT	*cis*-acting element involved in salicylic acid responsiveness
7	MYB recognition site	1	CCGTTG	MYB transcription factor binding site
8	CAAT-box	25	CAAAT/CCAAT/CAAT	common *cis*-acting element in promoter and enhancer regions
9	CARE	1	CAACTCAC	Unknown function
10	LAMP-element	1	CTTTATCA	part of a light responsive element
11	AAGAA-motif	2	GAAAGAA/gGTAAAGAAA	Unknown function
12	AT~TATA-box	2	TATATA	Unknown function
13	WRE3	1	CCACCT	Unknown function
14	MYC	2	CATTTG	MYC transcription factor binding site
15	ATCT-motif	1	AATCTAATCC	part of a conserved DNA module involved in light responsiveness
16	W box	1	TTGACC	*cis*-acting element involved in stress responsiveness
17	TATA-box	29	TATAAA/TATAA/TATA/ATTATA/ATATAT/TATAAATA/TATAAAT/TATATAA/TATATA/ATATAA/TATACA	core promoter element around -30 of transcription start
18	Myb-binding site	1	CAACAG	MYB transcription factor binding site
19	Box 4	1	ATTAAT	part of a conserved DNA module involved in light responsiveness
20	STRE	2	AGGGG	*cis*-acting regulatory element involved in STRE(CCCCT/AGGGG) abiotic stress
21	chs-CMA1a	1	TTACTTAA	part of a light responsive element
22	MRE	1	AACCTAA	MYB binding site involved in light responsiveness
23	GT1-motif	1	GGTTAA	light responsive element

## Data Availability

Not applicable.

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
