# Peer review of "Full-Length Transcriptome Characterization and Functional Analysis of Pathogenesis-Related Proteins in Lilium Oriental Hybrid ‘Sorbonne’ Infected with Botrytis elliptica"

_ijms, 2022, doi:10.3390/ijms24010425_

Round 1

Reviewer 1 Report

The manuscript “Transcriptome characterization and functional analysis of Pathogenesis-related proteins in Lilium oriental hybrid ‘Sorbonne’ infected with Botrytis elliptica” by Du et al. combines several (2-3, not sure) RNA-seq approaches without making clear where the various data is coming from.

It starts seemingly straight forward as a long-read RNA-seq approach based on PacBio SMRT sequencing. This part is described in some detail, though some clarification is required. The abstract mentions qRT-PCR results done on several pathogenesis-related genes confirming the transcriptome results. This certainly gave me as reader the impression that this was a “comparative transcriptomics” approach. But while reading the results, I started to  wonder about the lack of detail on Botrytis infection, and instead much detail on items not directly connect to Botrytis resistance, such as transcriptome statistics. When I finally reached the expected differential gene expression it dawned on me that this part is from “other transcriptomics data” not clearly explained or described; supplementary table S3 states: “Information on the nine highly expressed PRs identified from our transcriptome databases in this study”. What does this mean; where does this data come from? I realized eventually that it is from another study; that is fine, but state this more clearly. And make an effort to connect both RNA-seq approaches. What is the advantage of the current SMRT sequencing, and how does it enable new/improved analysis of the previous transcriptome data? 

Experimentally, the approaches appear sound, and the new full-length transcriptomics should indeed be useful, especially in light of a lacking genome sequence.

English language revisions needed throughout. This paper is difficult to understand for the reader. Abbreviations are often not spelled out at first mentioning. Better explanation of PacBio SMRT sequencing needed. “The PacBio sequencing platform is a circular sequencing system. The high-quality sequencing reads produced by single molecules during sequencing are called polymerase reads.” This really doesn’t help the reader understand what was done.

At the same time, it is important to find a balance between what the reader needs to know to understand the paper, and what should be obvious. For example, there is no need to explain what a transcription factor is (“Transcription factors (TFs) are protein molecules that bind to specific cis-acting elements upstream of the 5' end of a target gene”).

Additional points

Results

The significantly enriched pathways associated with the transcripts”

Enriched in comparison to what? 

“lncRNAs were qualitatively analyzed using four software: CNCI, CPAT, CPC, and Pfam.” What does that mean? Distinguish between software and databases; I know Pfam as a database; what are the others and how/to what end were those used?

“Within 12 hours of B. elliptica inoculation, the expression levels of the other 8 genes did not change significantly, except for LhSorPR10-1, which was significantly induced after 6 hours of pathogen infection in Sorbonne.”

When comparing the change from 6h control to 6h infected, it seems that PR genes show an increase in expression, from blue (down, control) to yellow/orange (no longer down); though it is odd that the control shows downregulation. What does that mean; expression is down after 6 hours in the non-infected control? Were these done in biological replicates; how many?

“The hormone-induced expression pattern of LhSorPR4-2 under pathogen-related phytohormones treatment, including SA, MeJA, and ETH, was analyzed (Figure 6). “

Sounds good, but  give the reader a brief  rationale for testing the effect of various plant hormones.

“A plant vector was constructed whereby translation of ORFs was modulated by a ZmUbi2 promoter,”

Well, promoters control transcription; perhaps replace “translation of ORFs” with “gene expression”? Also, tell the reader what type of expression is to expect from this promoter; is this a constitutive promoter that should be active in all tissues? 

Discussion

PacBio Iso-Seq was used, but how are these findings useful? Was there any analysis of alternative splice forms? As far as I can tell, the exploration of resistance was based on other (not-described) transcriptome sequencing.

Address in the discussion how the PacBio Iso-Seq is useful in the search to identify the mechanism of Botrytisresistance. And find a way to better connect it to the rest of the paper, which seems to be based on other RNA-seq techniques.

“In this study, overexpression of LhSorPR4-2 gene in Lilium oriental hybrid ‘Sorbonne’ was delayed compared with that of wild-type plants”

What does that mean, overexpression was delayed? Did you really mean that, or were you intending to say something else here? This draft need careful revisions.

Materials and Methods.

Here I found some information on the connection between both sequencing approaches.: Highly expressed PR genes were obtained from the second-generation transcriptome and combined with third-generation  transcriptome data to obtain full-length sequences.”

Instead of “the second-generation”, you could state “previously obtained second-generation…(citation)” ; also make this clear in abstract and results, and supplementary table S3

Conclusion

I liked the connection that was made here between the various components of the paper: “These findings lay a foundation for cloning of the full-length genes, gene function analysis, and elucidation of the synthesis mechanism of active components against B. elliptica in ‘Sorbonne’. “ More explanations like this within the rest of the paper would help in merging the various components into a cohesive story.

Author Response

Dear review 1,
         We appreciate your rigorous attitude, professional evaluation, and constructive suggestions. Thank you for spending a lot of valuable time patiently reviewing manuscripts. We have solved the problems you raised one by one. With best wishes for happiness in your life and work. Please see the attachment.
With kindest regards,
Dr. Daofeng Liu

Reviewer 2 Report

The paper applied updated biotechnology to characterize and analyze functional genes which undertake pathogen resistance of Lilium 'Sorbonne". The data was well organized and the obtained data was sufficiently discussed and interpreted. Some minor points should be revised before publication.

1. English presentation is understandable, but it should be polished by a native English speaker to remove grammaticals and typos.

2. Line 152-157: This part should be in the materials and methods.

3. Line 185: Please correct the subheading. It should be part 2.4 not 3.4. Please also correct the following parts.

4. Line 222: It should be " In order to verify...."

5. Line 443: It should be part 4.4 not 4.5. Please also correct the following parts

Author Response

Dear review 2,
        Thank you for your recognition of our work,kind comments and helpful suggestions. We have solved the problem you raised one by one. May happiness and health be with you always. Please see the attachment
Yours Sincerely
Dr. Daofeng Liu

Round 2

Reviewer 1 Report

Authors addressed my concerns; I believe the manuscript is now ready for publication